# Bias Correction of Learned Generative Models using Likelihood-Free Importance Weighting

**Aditya Grover**[1], **Jiaming Song**[1], **Alekh Agarwal**[2], **Kenneth Tran**[2],
**Ashish Kapoor**[2], **Eric Horvitz**[2], **Stefano Ermon**[1]
[1]Stanford University, [2]Microsoft Research, Redmond

## Abstract

A learned generative model often produces biased statistics relative to the underlying data distribution. A standard technique to correct this bias is importance sampling, where samples from the model are weighted by the likelihood ratio under model and true distributions. When the likelihood ratio is unknown, it can be estimated by training a probabilistic classifier to distinguish samples from the two distributions. We show that this likelihood-free importance weighting method induces a new energy-based model and employ it to correct for the bias in existing models. We find that this technique consistently improves standard goodness-of-fit metrics for evaluating the sample quality of state-of-the-art deep generative models, suggesting reduced bias. Finally, we demonstrate its utility on representative applications in a) data augmentation for classification using generative adversarial networks, and b) model-based policy evaluation using off-policy data.

## 1 Introduction

Learning generative models of complex environments from high-dimensional observations is a long-standing challenge in machine learning. Once learned, these models are used to draw inferences and to plan future actions. For example, in data augmentation, samples from a learned model are used to enrich a dataset for supervised learning [1]. In model-based off-policy policy evaluation (henceforth MBOPE), a learned dynamics model is used to simulate and evaluate a target policy without real-world deployment [2], which is especially valuable for risk-sensitive applications [3]. In spite of the recent successes of deep generative models, existing theoretical results show that learning distributions in an unbiased manner is either impossible or has prohibitive sample complexity [4, 5]. Consequently, the models used in practice are inherently *biased*,[1] and can lead to misleading downstream inferences.

In order to address this issue, we start from the observation that many typical uses of generative models involve computing expectations under the model. For instance, in MBOPE, we seek to find the expected return of a policy under a trajectory distribution defined by this policy and a learned dynamics model. A classical recipe for correcting the bias in expectations, when samples from a different distribution than the ground truth are available, is to importance weight the samples according to the likelihood ratio [6]. If the importance weights were exact, the resulting estimates are unbiased. But in practice, the likelihood ratio is unknown and needs to be estimated since the true data distribution is unknown and even the model likelihood is intractable or ill-defined for many deep generative models, e.g., variational autoencoders [7] and generative adversarial networks [8].

Our proposed solution to estimate the importance weights is to train a calibrated, probabilistic classifier to distinguish samples from the data distribution and the generative model. As shown in prior work, the output of such classifiers can be used to extract density ratios [9]. Appealingly, this estimation procedure is likelihood-free since it only requires samples from the two distributions.

Together, the generative model and the importance weighting function (specified via a binary classifier) induce a new energy function. While exact density estimation and sampling from this induced energy-based model is intractable, we can derive a particle based approximation which permits efficient sampling via resampling based methods. We derive conditions on the quality of the weighting function such that the induced model provably improves the fit to the the data distribution.

Empirically, we evaluate our bias reduction framework on three main sets of experiments. First, we consider goodness-of-fit metrics for evaluating sample quality metrics of a likelihood-based and a likelihood-free state-of-the-art (SOTA) model on the CIFAR-10 dataset. All these metrics are defined as Monte Carlo estimates from the generated samples. By importance weighting samples, we observe a bias reduction of 23.35% and 13.48% averaged across commonly used sample quality metrics on PixelCNN++ [10] and SNGAN [11] models respectively.

Next, we demonstrate the utility of our approach on the task of data augmentation for multi-class classification on the Omniglot dataset [12]. We show that, while naively extending the model with samples from a data augmentation, a generative adversarial network [1] is not very effective for multi-class classification, we can improve classification accuracy from 66.03% to 68.18% by importance weighting the contributions of each augmented data point.

Finally, we demonstrate bias reduction for MBOPE [13]. A typical MBOPE approach is to first estimate a generative model of the dynamics using off-policy data and then evaluate the policy via Monte Carlo [2, 14]. Again, we observe that correcting the bias of the estimated dynamics model via importance weighting reduces RMSE for MBOPE by 50.25% on 3 MuJoCo environments [15].

## 2   Preliminaries

**Notation.** Unless explicitly stated otherwise, we assume that probability distributions admit absolutely continuous densities on a suitable reference measure. We use uppercase notation $X, Y, Z$ to denote random variables and lowercase notation $x, y, z$ to denote specific values in the corresponding sample spaces $\mathcal{X}, \mathcal{Y}, \mathcal{Z}$. We use boldface for multivariate random variables and their vector values.

**Background.** Consider a finite dataset $D_{\text{train}}$ of instances $\mathbf{x}$ drawn i.i.d. from a fixed (unknown) distribution $p_{\text{data}}$. Given $D_{\text{train}}$, the goal of generative modeling is to learn a distribution $p_\theta$ to approximate $p_{\text{data}}$. Here, $\theta$ denotes the model parameters, e.g. weights in a neural network for deep generative models. The parameters can be learned via maximum likelihood estimation (MLE) as in the case of autoregressive models [16], normalizing flows [17], and variational autoencoders [7, 18], or via adversarial training e.g., using generative adversarial networks [8, 19] and variants.

**Monte Carlo Evaluation** We are interested in use cases where the goal is to evaluate or optimize expectations of functions under some distribution $p$ (either equal or close to the data distribution $p_{\text{data}}$). Assuming access to samples from $p$ as well some generative model $p_\theta$, one extreme is to evaluate the sample average using the samples from $p$ alone. However, this ignores the availability of $p_\theta$, through which we have a virtually unlimited access of generated samples ignoring computational constraints and hence, could improve the accuracy of our estimates when $p_\theta$ is close to $p$. We begin by presenting a direct motivating use case of data augmentation using generative models for training classifiers which generalize better.

**Example Use Case:** Sufficient labeled training data for learning classification and regression system is often expensive to obtain or susceptible to noise. *Data augmentation* seeks to overcome this shortcoming by artificially injecting new datapoints into the training set. These new datapoints are derived from an existing labeled dataset, either by manual transformations (e.g., rotations, flips for images), or alternatively, learned via a generative model [1, 20].

Consider a supervised learning task over a labeled dataset $D_{\text{cl}}$. The dataset consists of feature and label pairs $(\mathbf{x}, \mathbf{y})$, each of which is assumed to be sampled independently from a data distribution $p_{\text{data}}(\mathbf{x}, \mathbf{y})$ defined over $\mathcal{X} \times \mathcal{Y}$. Further, let $\mathcal{Y} \subseteq \mathbb{R}^k$. In order to learn a classifier $f_\psi : \mathcal{X} \rightarrow \mathbb{R}^k$ with parameters $\psi$, we minimize the expectation of a loss $\ell : \mathcal{Y} \times \mathbb{R}^k \rightarrow \mathbb{R}$ over the dataset $D_{\text{cl}}$:

$$\mathbb{E}_{p_{\text{data}}(\mathbf{x}, \mathbf{y})}[\ell(\mathbf{y}, f_\psi(\mathbf{x}))] \approx \frac{1}{|D_{\text{cl}}|} \sum_{(\mathbf{x}, \mathbf{y}) \sim D_{\text{cl}}} \ell(\mathbf{y}, f_\psi(\mathbf{x})). \tag{1}$$

E.g., $\ell$ could be the cross-entropy loss. A generative model for the task of data augmentation learns a joint distribution $p_\theta(\mathbf{x}, \mathbf{y})$. Several algorithmic variants exist for learning the model's joint distribution and we defer the specifics to the experiments section. Once the generative model is learned, it can be used to optimize the expected classification loss in Eq. (1) under a mixture distribution of empirical data distributions and generative model distributions given as:

$$p_{\mathrm{mix}}(\mathbf{x}, \mathbf{y}) = mp_{\mathrm{data}}(\mathbf{x}, \mathbf{y}) + (1 - m)p_\theta(\mathbf{x}, \mathbf{y}) \qquad (2)$$

for a suitable choice of the mixture weights $m \in [0, 1]$. Notice that, while the eventual task here is optimization, reliably evaluating the expected loss of a candidate parameter $\psi$ is an important ingredient. We focus on this basic question first in advance of leveraging the solution for data augmentation. Further, even if evaluating the expectation once is easy, optimization requires us to do repeated evaluation (for different values of $\psi$) which is significantly more challenging. Also observe that the distribution $p$ under which we seek expectations is same as $p_{\mathrm{data}}$ here, and we rely on the generalization of $p_\theta$ to generate transformations of an instance in the dataset which are not explicitly present, but plausibly observed in other, similar instances [21].

## 3   Likelihood-Free Importance Weighting

Whenever the distribution $p$, under which we seek expectations, differs from $p_\theta$, model-based estimates exhibit bias. In this section, we start out by formalizing bias for Monte Carlo expectations and subsequently propose a bias reduction strategy based on likelihood-free importance weighting (LFIW). We are interested in evaluating expectations of a class of functions of interest $f \in \mathcal{F}$ w.r.t. the distribution $p$. For any given $f : \mathcal{X} \to \mathbb{R}$, we have $\mathbb{E}_{\mathbf{x} \sim p}[f(\mathbf{x})] = \int p(\mathbf{x})f(\mathbf{x})\mathrm{d}\mathbf{x}$.

Given access to samples from a generative model $p_\theta$, if we knew the densities for both $p$ and $p_\theta$, then a classical scheme to evaluate expectations under $p$ using samples from $p_\theta$ is to use importance sampling [6]. We reweight each sample from $p_\theta$ according to its likelihood ratio under $p$ and $p_\theta$ and compute a weighted average of the function $f$ over these samples.

$$\mathbb{E}_{\mathbf{x} \sim p}[f(\mathbf{x})] = \mathbb{E}_{\mathbf{x} \sim p_\theta}\left[\frac{p(\mathbf{x})}{p_\theta(\mathbf{x})}f(\mathbf{x})\right] \approx \frac{1}{T}\sum_{i=1}^{T} w(\mathbf{x}_i)f(\mathbf{x}_i) \qquad (3)$$

where $w(\mathbf{x}_i) := {p(\mathbf{x}_i)}/{p_\theta(\mathbf{x}_i)}$ is the importance weight for $\mathbf{x}_i \sim p_\theta$. The validity of this procedure is subject to the use of a proposal $p_\theta$ such that for all $\mathbf{x} \in \mathcal{X}$ where $p_\theta(\mathbf{x}) = 0$, we also have $f(\mathbf{x})p(\mathbf{x}) = 0$.[2]

To apply this technique to reduce the bias of a generative sampler $p_\theta$ w.r.t. $p$, we require knowledge of the importance weights $w(\mathbf{x})$ for any $\mathbf{x} \sim p_\theta$. However, we typically only have a sampling access to $p$ via finite datasets. For instance, in the data augmentation example above, where $p = p_{\mathrm{data}}$, the unknown distribution used to learn $p_\theta$. Hence we need a scheme to learn the weights $w(\mathbf{x})$, using samples from $p$ and $p_\theta$, which is the problem we tackle next. In order to do this, we consider a binary classification problem over $\mathcal{X} \times \mathcal{Y}$ where $\mathcal{Y} = \{0, 1\}$ and the joint distribution is denoted as $q(\mathbf{x}, y)$. Let $\gamma = \frac{q(y=0)}{q(y=1)} > 0$ denote any fixed odds ratio. To specify the joint $q(\mathbf{x}, y)$, we additionally need the conditional $q(\mathbf{x}|y)$ which we define as follows:

$$q(\mathbf{x}|y) = \begin{cases} p_\theta(\mathbf{x}) \text{ if } y = 0 \\ p(\mathbf{x}) \text{ otherwise.} \end{cases} \qquad (4)$$

Since we only assume sample access to $p$ and $p_\theta(\mathbf{x})$, our strategy would be to estimate the conditional above via *learning* a probabilistic binary classifier. To train the classifier, we only require datasets of samples from $p_\theta(\mathbf{x})$ and $p(\mathbf{x})$ and estimate $\gamma$ to be the ratio of the size of two datasets. Let $c_\phi : \mathcal{X} \to [0, 1]$ denote the probability assigned by the classifier with parameters $\phi$ to a sample $\mathbf{x}$ belonging to the positive class $y = 1$. As shown in prior work [9, 22], if $c_\phi$ is Bayes optimal, then the importance weights can be obtained via this classifier as:

$$w_\phi(\mathbf{x}) = \frac{p(\mathbf{x})}{p_\theta(\mathbf{x})} = \gamma \frac{c_\phi(\mathbf{x})}{1 - c_\phi(\mathbf{x})}. \qquad (5)$$

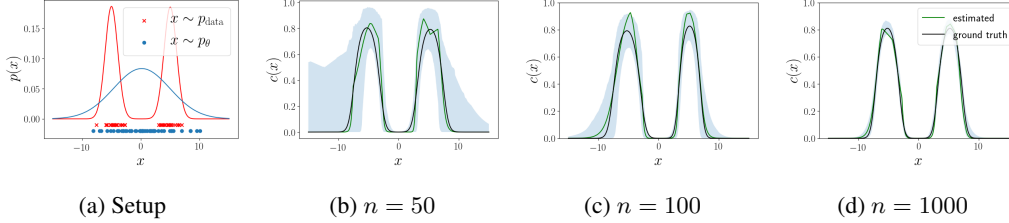

(a) Setup        (b) $n = 50$        (c) $n = 100$        (d) $n = 1000$

Figure 1: Importance Weight Estimation using Probabilistic Classifiers. (a) A univariate Gaussian (blue) is fit to samples from a mixture of two Gaussians (red). (b-d) Estimated class probabilities (with 95% confidence intervals based on 1000 bootstraps) for varying number of points $n$, where $n$ is the number of points used for training the generative model and multilayer perceptron.

In practice, we do not have access to a Bayes optimal classifier and hence, the estimated importance weights will not be exact. Consequently, we can hope to reduce the bias as opposed to eliminating it entirely. Hence, our default LFIW estimator is given as:

$$\mathbb{E}_{\mathbf{x}\sim p}[f(\mathbf{x})] \approx \frac{1}{T}\sum_{i=1}^{T}\hat{w}_\phi(\mathbf{x}_i)f(\mathbf{x}_i) \tag{6}$$

where $\hat{w}_\phi(\mathbf{x}_i) = \gamma\frac{c_\phi(\mathbf{x}_i)}{1-c_\phi(\mathbf{x}_i)}$ is the importance weight for $\mathbf{x}_i \sim p_\theta$ estimated via $c_\phi(\mathbf{x})$.

**Practical Considerations.** Besides imperfections in the classifier, the quality of a generative model also dictates the efficacy of importance weighting. For example, images generated by deep generative models often possess distinct artifacts which can be exploited by the classifier to give highly-confident predictions [23, 24]. This could lead to very small importance weights for some generated images, and consequently greater relative variance in the importance weights across the Monte Carlo batch. Below, we present some practical variants of LFIW estimator to offset this challenge.

1. *Self-normalization:* The self-normalized LFIW estimator for Monte Carlo evaluation normalizes the importance weights across a sampled batch:

$$\mathbb{E}_{\mathbf{x}\sim p}[f(\mathbf{x})] \approx \sum_{i=1}^{T}\frac{\hat{w}_\phi(\mathbf{x}_i)}{\sum_{j=1}^{T}\hat{w}_\phi(\mathbf{x}_j)}f(\mathbf{x}_i) \text{ where } \mathbf{x}_i \sim p_\theta. \tag{7}$$

2. *Flattening:* The flattened LFIW estimator interpolates between the uniform importance weights and the default LFIW weights via a power scaling parameter $\alpha \geq 0$:

$$\mathbb{E}_{\mathbf{x}\sim p}[f(\mathbf{x})] \approx \frac{1}{T}\sum_{i=1}^{T}\hat{w}_\phi(\mathbf{x}_i)^\alpha f(\mathbf{x}_i) \text{ where } \mathbf{x}_i \sim p_\theta. \tag{8}$$

For $\alpha = 0$, there is no bias correction, and $\alpha = 1$ returns the default estimator in Eq. (6). For intermediate values of $\alpha$, we can trade-off bias reduction with any undesirable variance introduced.

3. *Clipping:* The clipped LFIW estimator specifies a lower bound $\beta \geq 0$ on the importance weights:

$$\mathbb{E}_{\mathbf{x}\sim p}[f(\mathbf{x})] \approx \frac{1}{T}\sum_{i=1}^{T}\max(\hat{w}_\phi(\mathbf{x}_i), \beta)f(\mathbf{x}_i) \text{ where } \mathbf{x}_i \sim p_\theta. \tag{9}$$

When $\beta = 0$, we recover the default LFIW estimator in Eq. (6). Finally, we note that these estimators are not exclusive and can be combined e.g., flattened or clipped weights can be normalized.

**Confidence intervals.** Since we have real and generated data coming from a finite dataset and parametric model respectively, we propose a combination of empirical and parametric bootstraps to derive confidence intervals around the estimated importance weights. See Appendix A for details.

**Synthetic experiment.** We visually illustrate our importance weighting approach in a toy experiment (Figure 1a). We are given a finite set of samples drawn from a mixture of two Gaussians (red). The model family is a unimodal Gaussian, illustrating mismatch due to a parametric model. The mean

---

**Algorithm 1** SIR for the Importance Resampled Energy-Based Model $p_{\theta,\phi}$

---

    **Input:** Generative Model $p_\theta$, Importance Weight Estimator $\hat{w}_\phi$, budget $T$

1: Sample $\mathbf{x}_1, \mathbf{x}_2, \ldots, \mathbf{x}_T$ independently from $p_\theta$
2: Estimate importance weights $\hat{w}(\mathbf{x}_1), \hat{w}(\mathbf{x}_2), \ldots, \hat{w}(\mathbf{x}_T)$
3: Compute $\hat{Z} \leftarrow \sum_{t=1}^{T} \hat{w}(\mathbf{x}_t)$
4: Sample $j \sim \text{Categorical}\left(\frac{\hat{w}(\mathbf{x}_1)}{\hat{Z}}, \frac{\hat{w}(\mathbf{x}_2)}{\hat{Z}}, \ldots, \frac{\hat{w}(\mathbf{x}_T)}{\hat{Z}}\right)$
5: **return** $\mathbf{x}_j$

---

and variance of the model are estimated by the empirical means and variances of the observed data. Using estimated model parameters, we then draw samples from the model (blue).

In Figure 1b, we show the probability assigned by a binary classifier to a point to be from true data distribution. Here, the classifier is a single hidden-layer multi-layer perceptron. The classifier is not Bayes optimal, which can be seen by the gaps between the optimal probabilities curve (black) and the estimated class probability curve (green). However, as we increase the number of real and generated examples $n$ in Figures 1c-d, the classifier approaches optimality. Furthermore, even its uncertainty shrinks with increasing data, as expected. In summary, this experiment demonstrates how a binary classifier can mitigate this bias due to a mismatched generative model.

## 4   Importance Resampled Energy-Based Model

In the previous section, we described a procedure to augment any base generative model $p_\theta$ with an importance weighting estimator $\hat{w}_\phi$ for debiased Monte Carlo evaluation. Here, we will use this augmentation to induce an *importance resampled energy-based model* with density $p_{\theta,\phi}$ given as:

$$p_{\theta,\phi}(\mathbf{x}) \propto p_\theta(\mathbf{x})\hat{w}_\phi(\mathbf{x}) \tag{10}$$

where the partition function is expressed as $Z_{\theta,\phi} = \int p_\theta(\mathbf{x})\hat{w}_\phi(\mathbf{x})\mathrm{d}\mathbf{x} = \mathbb{E}_{p_\theta}[\hat{w}_\phi(\mathbf{x})]$.

**Density Estimation.** Exact density estimation requires a handle on the density of the base model $p_\theta$ (typically intractable for models such as VAEs and GANs) and estimates of the partition function. Exactly computing the partition function is intractable. If $p_\theta$ permits fast sampling and importance weights are estimated via LFIW (requiring only a forward pass through the classifier network), we can obtain unbiased estimates via a Monte Carlo average, i.e., $Z_{\theta,\phi} \approx \frac{1}{T}\sum_{i=1}^{T}\hat{w}_\phi(\mathbf{x}_i)$ where $\mathbf{x}_i \sim p_\theta$. To reduce the variance, a potentially large number of samples are required. Since samples are obtained independently, the terms in the Monte Carlo average can be evaluated in parallel.

**Sampling-Importance-Resampling.** While exact sampling from $p_{\theta,\phi}$ is intractable, we can instead perform sample from a particle-based approximation to $p_{\theta,\phi}$ via sampling-importance-resampling [25, 26] (SIR). We define the SIR approximation to $p_{\theta,\phi}$ via the following density:

$$p_{\theta,\phi}^{\text{SIR}}(\mathbf{x}; T) := \mathbb{E}_{\mathbf{x}_2, \mathbf{x}_3, \ldots, \mathbf{x}_T \sim p_\theta}\left[\frac{\hat{w}_\phi(\mathbf{x})}{\hat{w}_\phi(\mathbf{x}) + \sum_{i=2}^{T}\hat{w}_\phi(\mathbf{x}_i)}p_\theta(\mathbf{x})\right] \tag{11}$$

where $T > 0$ denotes the number of independent samples (or "particles"). For any finite $T$, sampling from $p_{\theta,\phi}^{\text{SIR}}$ is tractable, as summarized in Algorithm 1. Moreover, any expectation w.r.t. the SIR approximation to the induced distribution can be evaluated in closed-form using the self-normalized LFIW estimator (Eq. 7). In the limit of $T \to \infty$, we recover the induced distribution $p_{\theta,\phi}$:

$$\lim_{T \to \infty} p_{\theta,\phi}^{\text{SIR}}(\mathbf{x}; T) = p_{\theta,\phi}(\mathbf{x}) \quad \forall \mathbf{x} \tag{12}$$

Next, we analyze conditions under which the resampled density $p_{\theta,\phi}$ provably improves the model fit to $p_{\text{data}}$. In order to do so, we further assume that $p_{\text{data}}$ is absolutely continuous w.r.t. $p_\theta$ and $p_{\theta,\phi}$. We define the change in KL via the importance resampled density as:

$$\Delta(p_{\text{data}}, p_\theta, p_{\theta,\phi}) := D_{\text{KL}}(p_{\text{data}}, p_{\theta,\phi}) - D_{\text{KL}}(p_{\text{data}}, p_\theta). \tag{13}$$

Substituting Eq. 10 in Eq. 13, we can simplify the above quantity as:

$$\Delta(p_{\text{data}}, p_\theta, p_{\theta,\phi}) = \mathbb{E}_{\mathbf{x} \sim p_{\text{data}}}[-\log(p_\theta(\mathbf{x})\hat{w}_\phi(\mathbf{x})) + \log Z_{\theta,\phi} + \log p_\theta(\mathbf{x})] \tag{14}$$

$$= \mathbb{E}_{\mathbf{x} \sim p_{\text{data}}}[\log \hat{w}_\phi(\mathbf{x})] - \log \mathbb{E}_{\mathbf{x} \sim p_\theta}[\hat{w}_\phi(\mathbf{x})]. \tag{15}$$

Table 1: Goodness-of-fit evaluation on CIFAR-10 dataset for PixelCNN++ and SNGAN. Standard errors computed over 10 runs. **Higher IS is better. Lower FID and KID scores are better.**

| Model | Evaluation | IS ($\uparrow$) | FID ($\downarrow$) | KID ($\downarrow$) |
|---|---|---|---|---|
| - | Reference | $11.09 \pm 0.1263$ | $5.20 \pm 0.0533$ | $0.008 \pm 0.0004$ |
| PixelCNN++ | Default (no debiasing) | $5.16 \pm 0.0117$ | $58.70 \pm 0.0506$ | $0.196 \pm 0.0001$ |
| | LFIW | $\mathbf{6.68} \pm 0.0773$ | $\mathbf{55.83} \pm 0.9695$ | $\mathbf{0.126} \pm 0.0009$ |
| SNGAN | Default (no debiasing) | $8.33 \pm 0.0280$ | $20.40 \pm 0.0747$ | $0.094 \pm 0.0002$ |
| | LFIW | $\mathbf{8.57} \pm 0.0325$ | $\mathbf{17.29} \pm 0.0698$ | $\mathbf{0.073} \pm 0.0004$ |

The above expression provides a necessary and sufficient condition for any positive real valued function (such as the LFIW classifier in Section 3) to improve the KL divergence fit to the underlying data distribution. In practice, an unbiased estimate of the LHS can be obtained via Monte Carlo averaging of log- importance weights based on $D_{\text{train}}$. The empirical estimate for the RHS is however biased.[3] To remedy this shortcoming, we consider the following necessary but insufficient condition.

**Proposition 1.** *If* $\Delta(p_{\text{data}}, p_\theta, p_{\theta,\phi}) \geq 0$*, then the following conditions hold:*

$$\mathbb{E}_{\mathbf{x} \sim p_{\text{data}}}[\hat{w}_\phi(\mathbf{x})] \geq \mathbb{E}_{\mathbf{x} \sim p_\theta}[\hat{w}_\phi(\mathbf{x})], \tag{16}$$

$$\mathbb{E}_{\mathbf{x} \sim p_{\text{data}}}[\log \hat{w}_\phi(\mathbf{x})] \geq \mathbb{E}_{\mathbf{x} \sim p_\theta}[\log \hat{w}_\phi(\mathbf{x})]. \tag{17}$$

The conditions in Eq. 16 and Eq. 17 follow directly via Jensen's inequality applied to the LHS and RHS of Eq. 15 respectively. Here, we note that estimates for the expectations in Eqs. 16-17 based on Monte Carlo averaging of (log-) importance weights are unbiased.

## 5   Application Use Cases

In all our experiments, the binary classifier for estimating the importance weights was a calibrated deep neural network trained to minimize the cross-entropy loss. The self-normalized LFIW in Eq. (7) worked best. Additional analysis on the estimators and experiment details are in Appendices B and C.

### 5.1   Goodness-of-fit testing

In the first set of experiments, we highlight the benefits of importance weighting for a debiased evaluation of three popularly used sample quality metrics viz. Inception Scores (IS) [27], Frechet Inception Distance (FID) [28], and Kernel Inception Distance (KID) [29]. All these scores can be formally expressed as empirical expectations with respect to the model. For all these metrics, we can simulate the population level unbiased case as a "reference score" wherein we artificially set both the real and generated sets of samples used for evaluation as finite, disjoint sets derived from $p_{\text{data}}$.

We evaluate the three metrics for two state-of-the-art models trained on the CIFAR-10 dataset viz. an autoregressive model PixelCNN++ [10] learned via maximum likelihood estimation and a latent variable model SNGAN [11] learned via adversarial training. For evaluating each metric, we draw 10,000 samples from the model. In Table 1, we report the metrics with and without the LFIW bias correction. The consistent debiased evaluation of these metrics via self-normalized LFIW suggest that the SIR approximation to the importance resampled distribution (Eq. 11) is a better fit to $p_{\text{data}}$.

### 5.2   Data Augmentation for Multi-Class Classification

We consider data augmentation via Data Augmentation Generative Adversarial Networks (DA-GAN) [1]. While DAGAN was motivated by and evaluated for the task of meta-learning, it can also be applied for multi-class classification scenarios, which is the setting we consider here. We trained a DAGAN on the Omniglot dataset of handwritten characters [12]. The DAGAN training procedure is described in the Appendix. The dataset is particularly relevant because it contains 1600+ classes but only 20 examples from each class and hence, could potentially benefit from augmented data.

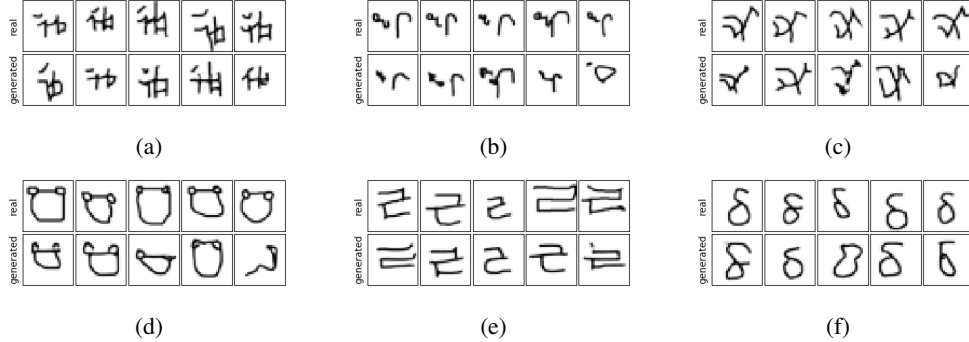

(a)          (b)          (c)

(d)          (e)          (f)

Figure 2: Qualitative evaluation of importance weighting for data augmentation. (a-f) Top row shows held-out data samples from a specific class in Omniglot. Bottom row shows generated samples from the same class *ranked in decreasing order* of importance weights.

Table 2: Classification accuracy on the Omniglot dataset. Standard errors computed over 5 runs.

| Dataset | $D_{\mathrm{cl}}$ | $D_{\mathrm{g}}$ | $D_{\mathrm{g}}$ w/ LFIW | $D_{\mathrm{cl}} + D_{\mathrm{g}}$ | $D_{\mathrm{cl}} + D_{\mathrm{g}}$ w/ LFIW |
|---|---|---|---|---|---|
| Accuracy | $0.6603 \pm 0.0012$ | $0.4431 \pm 0.0054$ | $0.4481 \pm 0.0056$ | $0.6600 \pm 0.0040$ | $\mathbf{0.6818} \pm 0.0022$ |

Once the model has been trained, it can be used for data augmentation in many ways. In particular, we consider ablation baselines that use various combinations of the real training data $D_{\mathrm{cl}}$ and generated data $D_{\mathrm{g}}$ for training a downstream classifier. When the generated data $D_{\mathrm{g}}$ is used, we can either use the data directly with uniform weighting for all training points, or choose to importance weight (LFIW) the contributions of the individual training points to the overall loss. The results are shown in Table 2. While generated data ($D_{\mathrm{g}}$) alone cannot be used to obtain competitive performance relative to the real data ($D_{\mathrm{cl}}$) on this task as expected, the bias it introduces for evaluation and subsequent optimization overshadows even the naive data augmentation ($D_{\mathrm{cl}} + D_{\mathrm{g}}$). In contrast, we can obtain significant improvements by importance weighting the generated points ($D_{\mathrm{cl}} + D_{\mathrm{g}}$ w/ LFIW).

Qualitatively, we can observe the effect of importance weighting in Figure 2. Here, we show true and generated samples for 6 randomly choosen classes (a-f) in the Omniglot dataset. The generated samples are ranked in decreasing order of the importance weights. There is no way to formally test the validity of such rankings and this criteria can also prefer points which have high density under $p_{\mathrm{data}}$ but are unlikely under $p_\theta$ since we are looking at ratios. Visual inspection suggests that the classifier is able to appropriately downweight poorer samples, as shown in Figure 2 (a, b, c, d - bottom right). There are also failure modes, such as the lowest ranked generated images in Figure 2 (e, f - bottom right) where the classifier weights reasonable generated samples poorly relative to others. This could be due to particular artifacts such as a tiny disconnected blurry speck in Figure 2 (e - bottom right) which could be more revealing to a classifier distinguishing real and generated data.

## 5.3 Model-based Off-policy Policy Evaluation

So far, we have seen use cases where the generative model was trained on data from the same distribution we wish to use for Monte Carlo evaluation. We can extend our debiasing framework to more involved settings when the generative model is a building block for specifying the full data generation process, e.g., trajectory data generated via a dynamics model along with an agent policy.

In particular, we consider the setting of off-policy policy evaluation (OPE), where the goal is to evaluate policies using experiences collected from a different policy. Formally, let $(\mathcal{S}, \mathcal{A}, r, P, \eta, T)$ denote an (undiscounted) Markov decision process with state space $\mathcal{S}$, action space $\mathcal{A}$, reward function $r$, transition $P$, initial state distribution $\eta$ and horizon $T$. Assume $\pi_e : \mathcal{S} \times \mathcal{A} \to [0, 1]$ is a known policy that we wish to evaluate. The probability of generating a certain trajectory $\tau = \{\mathbf{s}_0, \mathbf{a}_0, \mathbf{s}_1, \mathbf{a}_1, ..., \mathbf{s}_T, \mathbf{a}_T\}$ of length $T$ with policy $\pi_e$ and transition $P$ is given as:

$$p^\star(\tau) = \eta(\mathbf{s}_0) \prod_{t=0}^{T-1} \pi_e(\mathbf{a}_t|\mathbf{s}_t) P(\mathbf{s}_{t+1}|\mathbf{s}_t, \mathbf{a}_t). \tag{18}$$

The return on a trajectory $R(\tau)$ is the sum of the rewards across the state, action pairs in $\tau$: $R(\tau) = \sum_{t=1}^{T} r(\mathbf{s}_t, \mathbf{a}_t)$, where we assume a *known reward function* $r$.

Table 3: Off-policy policy evaluation on MuJoCo tasks. Standard error is over 10 Monte Carlo estimates where each estimate contains 100 randomly sampled trajectories.

| Environment | $v(\pi_e)$ (Ground truth) | $\tilde{v}(\pi_e)$ | $\hat{v}(\pi_e)$ (w/ LFIW) | $\hat{v}_{80}(\pi_e)$ (w/ LFIW) |
|---|---|---|---|---|
| Swimmer | $36.7 \pm 0.1$ | $100.4 \pm 3.2$ | $\mathbf{25.7 \pm 3.1}$ | $47.6 \pm 4.8$ |
| HalfCheetah | $241.7 \pm 3.56$ | $204.0 \pm 0.8$ | $217.8 \pm 4.0$ | $\mathbf{219.1 \pm 1.6}$ |
| HumanoidStandup | $14170 \pm 53$ | $8417 \pm 28$ | $\mathbf{9372 \pm 375}$ | $9221 \pm 381$ |

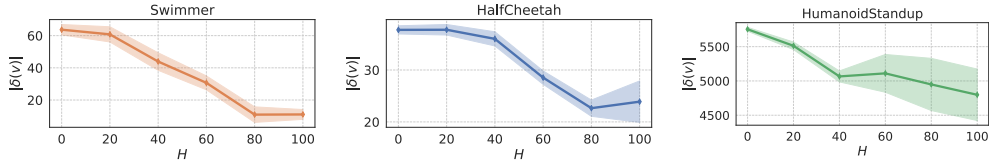

Figure 3: Estimation error $\delta(v) = v(\pi_e) - \hat{v}_H(\pi_e)$ for different values of $H$ (minimum 0, maximum 100). Shaded area denotes standard error over different random seeds.

We are interested in the value of a policy defined as $v(\pi_e) = \mathbb{E}_{\tau \sim p^*(\tau)}[R(\tau)]$. Evaluating $\pi_e$ requires the (unknown) transition dynamics $P$. The dynamics model is a conditional generative model of the next states $\mathbf{s}_{t+1}$ conditioned on the previous state-action pair $(\mathbf{s}_t, \mathbf{a}_t)$. If we have access to historical logged data $D_\tau$ of trajectories $\tau = \{\mathbf{s}_0, \mathbf{a}_0, \mathbf{s}_1, \mathbf{a}_1, \dots, \}$ from some behavioral policy $\pi_b : \mathcal{S} \times \mathcal{A} \to [0, 1]$, then we can use this off-policy data to train a dynamics model $P_\theta(\mathbf{s}_{t+1}|\mathbf{s}_t, \mathbf{a}_t)$. The policy $\pi_e$ can then be evaluated under this learned dynamics model as $\tilde{v}(\pi_e) = \mathbb{E}_{\tau \sim \tilde{p}(\tau)}[R(\tau)]$, where $\tilde{p}$ uses $P_\theta$ instead of the true dynamics in Eq. (18).

However, the trajectories sampled with $P_\theta$ could significantly deviate from samples from $P$ due to compounding errors [30]. In order to correct for this bias, we can use likelihood-free importance weighting on entire trajectories of data. The binary classifier $c(\mathbf{s}_t, \mathbf{a}_t, \mathbf{s}_{t+1})$ for estimating the importance weights in this case distinguishes between triples of true and generated transitions. For any true triple $(\mathbf{s}_t, \mathbf{a}_t, \mathbf{s}_{t+1})$ extracted from the off-policy data, the corresponding generated triple $(\mathbf{s}_t, \mathbf{a}_t, \hat{\mathbf{s}}_{t+1})$ only differs in the final transition state, i.e., $\hat{\mathbf{s}}_{t+1} \sim P_\theta(\hat{\mathbf{s}}_{t+1}|\mathbf{s}_t, \mathbf{a}_t)$. Such a classifier allows us to obtain the importance weights $\hat{w}(\mathbf{s}_t, \mathbf{a}_t, \hat{\mathbf{s}}_{t+1})$ for every predicted state transition $(\mathbf{s}_t, \mathbf{a}_t, \hat{\mathbf{s}}_{t+1})$. The importance weights for the trajectory $\tau$ can be derived from the importance weights of these individual transitions as:

$$\frac{p^\star(\tau)}{\tilde{p}(\tau)} = \frac{\prod_{t=0}^{T-1} P(\mathbf{s}_{t+1}|\mathbf{s}_t, \mathbf{a}_t)}{\prod_{t=0}^{T-1} P_\theta(\mathbf{s}_{t+1}|\mathbf{s}_t, \mathbf{a}_t)} = \prod_{t=0}^{T-1} \frac{P(\mathbf{s}_{t+1}|\mathbf{s}_t, \mathbf{a}_t)}{P_\theta(\mathbf{s}_{t+1}|\mathbf{s}_t, \mathbf{a}_t)} \approx \prod_{t=0}^{T-1} \hat{w}(\mathbf{s}_t, \mathbf{a}_t, \hat{\mathbf{s}}_{t+1}). \qquad (19)$$

Our final LFIW estimator is given as:

$$\hat{v}(\pi_e) = \mathbb{E}_{\tau \sim \tilde{p}(\tau)}\left[\prod_{t=0}^{T-1} \hat{w}(\mathbf{s}_t, \mathbf{a}_t, \hat{\mathbf{s}}_{t+1}) \cdot R(\tau)\right]. \qquad (20)$$

We consider three continuous control tasks in the MuJoCo simulator [15] from OpenAI gym [31] (in increasing number of state dimensions): Swimmer, HalfCheetah and HumanoidStandup. High dimensional state spaces makes it challenging to learning a reliable dynamics model in these environments. We train behavioral and evaluation policies using Proximal Policy Optimization [32] with different hyperparameters for the two policies. The dataset collected via trajectories from the behavior policy are used train a ensemble neural network dynamics model. We the use the trained dynamics model to evaluate $\tilde{v}(\pi_e)$ and its IW version $\hat{v}(\pi_e)$, and compare them with the ground truth returns $v(\pi_e)$. Each estimation is averaged over a set of 100 trajectories with horizon $T = 100$. Specifically, for $\hat{v}(\pi_e)$, we also average the estimation over 10 classifier instances trained with different random seeds on different trajectories. We further consider performing IW over only the first $H$ steps, and use uniform weights for the remainder, which we denote as $\hat{v}_H(\pi_e)$. This allow us to interpolate between $\tilde{v}(\pi_e) \equiv \hat{v}_0(\pi_e)$ and $\hat{v}(\pi_e) \equiv \hat{v}_T(\pi_e)$. Finally, as in the other experiments, we used the self-normalized variant (Eq. (7)) of the importance weighted estimator in Eq. (20).

We compare the policy evaluations under different environments in Table 3. These results show that the rewards estimated with the trained dynamics model differ from the ground truth by a large margin.

By importance weighting the trajectories, we obtain much more accurate policy evaluations. As expected, we also see that while LFIW leads to higher returns on average, the imbalance in trajectory importance weights due to the multiplicative weights of the state-action pairs can lead to higher variance in the importance weighted returns. In Figure 3, we demonstrate that policy evaluation becomes more accurate as more timesteps are used for LFIW evaluations, until around $80 - 100$ timesteps and thus empirically validates the benefits of importance weighting using a classifier. Given that our estimates have a large variance, it would be worthwhile to compose our approach with other variance reduction techniques such as (weighted) doubly robust estimation in future work [33], as well as incorporate these estimates within a framework such as MAGIC to further blend with model-free OPE [14]. In Appendix C.5.1, we also consider a stepwise LFIW estimator for MBOPE which applies importance weighting at the level of every decision as opposed to entire trajectories.

**Overall.** Across all our experiments, we observe that importance weighting the generated samples leads to uniformly better results, whether in terms of evaluating the quality of samples, or their utility in downstream tasks. Since the technique is a black-box wrapper around any generative model, we expect this to benefit a diverse set of tasks in follow-up works.

However, there is also some caution to be exercised with these techniques as evident from the results of Table 1. Note that in this table, the confidence intervals (computed using the reported standard errors) around the model scores after importance weighting still do not contain the reference scores obtained from the true model. This would not have been the case if our debiased estimator was completely unbiased and this observation reiterates our earlier claim that LFIW is reducing bias, as opposed to completely eliminating it. Indeed, when such a mismatch is observed, it is a good diagnostic to either learn more powerful classifiers to better approximate the Bayes optimum, or find additional data from $p_{\text{data}}$ in case the generative model fails the full support assumption.

# 6 Related Work & Discussion

Density ratios enjoy widespread use across machine learning e.g., for handling covariate shifts, class imbalance etc. [9, 34]. In generative modeling, estimating these ratios via binary classifiers is frequently used for defining learning objectives and two sample tests [19, 35, 35–41]. In particular, such classifiers have been used to define learning frameworks such as generative adversarial networks [8, 42], likelihood-free Approximate Bayesian Computation (ABC) [43] and earlier work in unsupervised-as-supervised learning [44] and noise contrastive estimation [43] among others. Recently, [45] used importance weighting to reweigh datapoints based on differences in training and test data distributions i.e., *dataset* bias. The key difference is that these works are explicitly interested in *learning* the parameters of a generative model. In contrast, we use the binary classifier for estimating importance weights to correct for the *model* bias of any *fixed* generative model.

Recent concurrent works [46–48] use MCMC and rejection sampling to explicitly transform or reject the generated samples. These methods require extra computation beyond training a classifier, in rejecting the samples or running Markov chains to convergence, unlike the proposed importance weighting strategy. For many model-based Monte Carlo evaluation usecases (e.g., data augmentation, MBOPE), this extra computation is unnecessary. If samples or density estimates are explicitly needed from the induced resampled distribution, we presented a particle-based approximation to the induced density where the number of particles is a tunable knob allowing for trading statistical accuracy with computational efficiency. Finally, we note resampling based techniques have been extensively studied in the context of improving variational approximations for latent variable generative models [49–52].

# 7 Conclusion

We identified bias with respect to a target data distribution as a fundamental challenge restricting the use of generative models as proposal distributions for Monte Carlo evaluation. We proposed a bias correction framework based on importance weighting. Here, any base generative model can be boosted with an importance weight estimator to induce an energy-based generative model. The importance weights are estimated in a likelihood-free fashion via a binary classifier. Empirically, we find the bias correction to be useful across a variety of tasks including goodness-of-fit sample quality tests, data augmentation, and off-policy policy evaluation. The ability to characterize the bias of a generative model is an important step towards using these models to guide decisions in high-stakes applications under uncertainty [53, 54], such as healthcare [55–57] and anomaly detection [58, 59].

## Acknowledgments

This project was initiated when AG was an intern at Microsoft Research. We are thankful to Daniel Levy, Rui Shu, Yang Song, and members of the Reinforcement Learning, Deep Learning, and Adaptive Systems and Interaction groups at Microsoft Research for helpful discussions and comments on early drafts. This research was supported by NSF (#1651565, #1522054, #1733686), ONR, AFOSR (FA9550-19-1-0024), and FLI.

## Footnotes

[1]We call a generative model biased if it produces biased statistics relative to the true data distribution.

[2]A stronger sufficient, but not necessary condition that is independent of $f$, states that the proposal $p_\theta$ is valid if it has a support larger than $p$, i.e., for all $\mathbf{x} \in \mathcal{X}$, $p_\theta(\mathbf{x}) = 0$ implies $p(\mathbf{x}) = 0$.

[3]If $\hat{Z}$ is an unbiased estimator for $Z$, then $\log \hat{Z}$ is a biased estimator for $\log Z$ via Jensen's inequality.

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
