[Supplementary Material]

## Appendices

## A Confidence Intervals via Bootstrap

Bootstrap is a widely-used tool in statistics for deriving confidence intervals by fitting ensembles of models on resampled data points. If the dataset is finite e.g., $D_{\text{train}}$, then the bootstrapped dataset is obtained via random sampling *with replacement* and confidence intervals are estimated via the *empirical bootstrap*. For a parametric model generating the dataset e.g., $p_\theta$, a fresh bootstrapped dataset is resampled from the model and confidence intervals are estimated via the *parametric bootstrap*. See [60] for a detailed review. In training a binary classifier, we can estimate the confidence intervals by retraining the classifier on a fresh sample of points from $p_\theta$ and a resampling of the training dataset $D_{\text{train}}$ (with replacement). Repeating this process over multiple runs and then taking a suitable quantile gives us the corresponding confidence intervals.

## B Bias-Variance of Different LFIW estimators

As discussed in Section 3, bias reduction using LFIW can suffer from issues where the importance weights are too small due to highly confident predictions of the binary classifier. Across a batch of Monte Carlo samples, this can increase the corresponding variance. Inspired from the importance sampling literature, we proposed additional mechanisms to mitigate this additional variance at the cost of reduced debiasing in Eqs. (7-9). We now look at the empirical bias-variance trade-off of these different estimators via a simple experiment below.

Our setup follows the goodness-of-fit testing experiments in Section 5. The statistics we choose to estimate is simply are the 2048 activations of the prefinal layer of the Inception Network, averaged across the test set of $10,000$ samples of CIFAR-10.

That is, the true statistics $\mathbf{s} = \{s_1, s_2, \cdots, s_{2048}\}$ are given by:

$$s_j = \frac{1}{|D_{\text{test}}|} \sum_{\mathbf{x} \in D_{\text{test}}} a_j(\mathbf{x}) \tag{21}$$

where $a_j$ is the $j$-th prefinal layer activation of the Inception Network. Note that set of statistics $\mathbf{s}$ is fixed (computed once on the test set).

To estimate these statistics, we will use different estimators. For example, the default estimator involving no reweighting is given as:

$$\hat{s}_j = \frac{1}{T} \sum_{i=1}^{T} a_j(\mathbf{x}) \tag{22}$$

where $\mathbf{x} \sim p_\theta$.

Note that $\hat{s}_j$ is a random variable since it depends on the $T$ samples drawn from $p_\theta$. Similar to Eq. (22), other variants of the LFIW estimators proposed in Section 3 can be derived using Eqs. (7-9). For any LFIW estimate $\hat{s}_j$, we can use the standard decomposition of the expected mean-squared error into terms corresponding to the (squared) bias and variance as shown below.

$$\mathbb{E}[(s_j - \hat{s}_j)^2] = s_j^2 - 2s_j \mathbb{E}[\hat{s}_j] + \mathbb{E}[\hat{s}_j]^2 \tag{23}$$

$$= s_j^2 - 2s_j \mathbb{E}[\hat{s}_j] + (\mathbb{E}[\hat{s}_j])^2 + \mathbb{E}[\hat{s}_j^2] - (\mathbb{E}[\hat{s}_j])^2 \tag{24}$$

$$= \underbrace{(s_j - \mathbb{E}[\hat{s}_j])^2}_{\text{Bias}^2} + \underbrace{\mathbb{E}[\hat{s}_j^2] - (\mathbb{E}[\hat{s}_j])^2}_{\text{Variance}}. \tag{25}$$

In Table 4, we report the bias and variance terms of the estimators averaged over 10 draws of $T = 10,0000$ samples and further averaging over all 2048 statistics corresponding to $\mathbf{s}$. We observe that self-normalization performs consistently well and is the best or second best in terms of bias and MSE in all cases. The flattened estimator with no debiasing (corresponding to $\alpha = 0$) has lower bias and higher variance than the self-normalized estimator. Amongst the flattening estimators, lower

Table 4: Bias-variance analysis for PixelCNN++ and SNGAN when $T = 10,000$. Standard errors over the absolute values of bias and variance evaluations are computed over the 2048 activation statistics. Lower absolute values of bias, lower variance, and lower MSE is better.

| Model | Evaluation | \|Bias\| ($\downarrow$) | Variance ($\downarrow$) | MSE ($\downarrow$) |
|---|---|---|---|---|
| PixelCNN++ | Self-norm | **0.0240** $\pm$ 0.0014 | 0.0002935 $\pm$ 7.22e-06 | **0.0046** $\pm$ 0.00031 |
| | Flattening ($\alpha = 0$) | 0.0330 $\pm$ 0.0023 | 9.1e-06 $\pm$ 2.6e-07 | 0.0116 $\pm$ 0.00093 |
| | Flattening ($\alpha = 0.25$) | 0.1042 $\pm$ 0.0018 | **5.1e-06** $\pm$ 1.5e-07 | 0.0175 $\pm$ 0.00138 |
| | Flattening ($\alpha = 0.5$) | 0.1545 $\pm$ 0.0022 | 8.4e-06 $\pm$ 3.7e-07 | 0.0335 $\pm$ 0.00246 |
| | Flattening ($\alpha = 0.75$) | 0.1626 $\pm$ 0.0022 | 3.19e-05 $\pm$ 2e-06 | 0.0364 $\pm$ 0.00259 |
| | Flattening ($\alpha = 1.0$) | 0.1359 $\pm$ 0.0018 | 0.0002344 $\pm$ 1.619e-05 | 0.0257 $\pm$ 0.00175 |
| | Clipping ($\beta = 0.001$) | 0.1359 $\pm$ 0.0018 | 0.0002344 $\pm$ 1.619e-05 | 0.0257 $\pm$ 0.00175 |
| | Clipping ($\beta = 0.01$) | 0.1357 $\pm$ 0.0018 | 0.0002343 $\pm$ 1.618e-05 | 0.0256 $\pm$ 0.00175 |
| | Clipping ($\beta = 0.1$) | 0.1233 $\pm$ 0.0017 | 0.000234 $\pm$ 1.611e-05 | 0.0215 $\pm$ 0.00149 |
| | Clipping ($\beta = 1.0$) | 0.1255 $\pm$ 0.0030 | 0.0002429 $\pm$ 1.606e-05 | 0.0340 $\pm$ 0.00230 |
| SNGAN | Self-norm | 0.0178 $\pm$ 0.0008 | 1.98e-05 $\pm$ 5.9e-07 | 0.0016 $\pm$ 0.00023 |
| | Flattening ($\alpha = 0$) | 0.0257 $\pm$ 0.0010 | 9.1e-06 $\pm$ 2.3e-07 | 0.0026 $\pm$ 0.00027 |
| | Flattening ($\alpha = 0.25$) | **0.0096** $\pm$ 0.0007 | **8.4e-06** $\pm$ 3.1e-07 | **0.0011** $\pm$ 8e-05 |
| | Flattening ($\alpha = 0.5$) | 0.0295 $\pm$ 0.0006 | 1.15e-05 $\pm$ 6.4e-07 | 0.0017 $\pm$ 0.00011 |
| | Flattening ($\alpha = 0.75$) | 0.0361 $\pm$ 0.0006 | 1.93e-05 $\pm$ 1.39e-06 | 0.002 $\pm$ 0.00012 |
| | Flattening ($\alpha = 1.0$) | 0.0297 $\pm$ 0.0005 | 3.76e-05 $\pm$ 3.08e-06 | 0.0015 $\pm$ 7e-05 |
| | Clipping ($\beta = 0.001$) | 0.0297 $\pm$ 0.0005 | 3.76e-05 $\pm$ 3.08e-06 | 0.0015 $\pm$ 7e-05 |
| | Clipping ($\beta = 0.01$) | 0.0297 $\pm$ 0.0005 | 3.76e-05 $\pm$ 3.08e-06 | 0.0015 $\pm$ 7e-05 |
| | Clipping ($\beta = 0.1$) | 0.0296 $\pm$ 0.0005 | 3.76e-05 $\pm$ 3.08e-06 | 0.0015 $\pm$ 7e-05 |
| | Clipping ($\beta = 1.0$) | 0.1002 $\pm$ 0.0018 | 3.03e-05 $\pm$ 2.18e-06 | 0.0170 $\pm$ 0.00171 |

values of $\alpha$ seem to provide the best bias-variance trade-off. The clipped estimators do not perform well in this setting, with lower values of $\beta$ slightly preferable over larger values. We repeat the same experiment with $T = 5,000$ samples and report the results in Table 5. While the variance increases as expected (by almost an order of magnitude), the estimator bias remains roughly the same.

Table 5: Bias-variance analysis for PixelCNN++ and SNGAN when $T = 5,000$. Standard errors over the absolute values of bias and variance evaluations are computed over the 2048 activation statistics. Lower absolute values of bias, lower variance, and lower MSE is better.

| Model | Evaluation | \|Bias\| ($\downarrow$) | Variance ($\downarrow$) | MSE ($\downarrow$) |
|---|---|---|---|---|
| PixelCNN++ | Self-norm | **0.023** $\pm$ 0.0014 | 0.0005086 $\pm$ 1.317e-05 | **0.0049** $\pm$ 0.00033 |
| | Flattening ($\alpha = 0$) | 0.0330 $\pm$ 0.0023 | **1.65e-05** $\pm$ 4.6e-07 | 0.0116 $\pm$ 0.00093 |
| | Flattening ($\alpha = 0.25$) | 0.1038 $\pm$ 0.0018 | 9.5e-06 $\pm$ 3e-07 | 0.0174 $\pm$ 0.00137 |
| | Flattening ($\alpha = 0.5$) | 0.1539 $\pm$ 0.0022 | 1.74e-05 $\pm$ 8e-07 | 0.0332 $\pm$ 0.00244 |
| | Flattening ($\alpha = 0.75$) | 0.1620 $\pm$ 0.0022 | 6.24e-05 $\pm$ 3.83e-06 | 0.0362 $\pm$ 0.00256 |
| | Flattening ($\alpha = 1.0$) | 0.1360 $\pm$ 0.0018 | 0.0003856 $\pm$ 2.615e-05 | 0.0258 $\pm$ 0.00174 |
| | Clipping ($\beta = 0.001$) | 0.1360 $\pm$ 0.0018 | 0.0003856 $\pm$ 2.615e-05 | 0.0258 $\pm$ 0.00174 |
| | Clipping ($\beta = 0.01$) | 0.1358 $\pm$ 0.0018 | 0.0003856 $\pm$ 2.615e-05 | 0.0257 $\pm$ 0.00173 |
| | Clipping ($\beta = 0.1$) | 0.1234 $\pm$ 0.0017 | 0.0003851 $\pm$ 2.599e-05 | 0.0217 $\pm$ 0.00148 |
| | Clipping ($\beta = 1.0$) | 0.1250 $\pm$ 0.0030 | 0.0003821 $\pm$ 2.376e-05 | 0.0341 $\pm$ 0.00232 |
| SNGAN | Self-norm | 0.0176 $\pm$ 0.0008 | 3.88e-05 $\pm$ 9.6e-07 | 0.0016 $\pm$ 0.00022 |
| | Flattening ($\alpha = 0$) | 0.0256 $\pm$ 0.0010 | 1.71e-05 $\pm$ 4.3e-07 | 0.0027 $\pm$ 0.00027 |
| | Flattening ($\alpha = 0.25$) | **0.0099** $\pm$ 0.0007 | **1.44e-05** $\pm$ 3.7e-07 | **0.0011** $\pm$ 8e-05 |
| | Flattening ($\alpha = 0.5$) | 0.0298 $\pm$ 0.0006 | 1.62e-05 $\pm$ 5.3e-07 | 0.0017 $\pm$ 0.00012 |
| | Flattening ($\alpha = 0.75$) | 0.0366 $\pm$ 0.0006 | 2.38e-05 $\pm$ 1.11e-06 | 0.0021 $\pm$ 0.00012 |
| | Flattening ($\alpha = 1.0$) | 0.0302 $\pm$ 0.0005 | 4.56e-05 $\pm$ 2.8e-06 | 0.0015 $\pm$ 7e-05 |
| | Clipping ($\beta = 0.001$) | 0.0302 $\pm$ 0.0005 | 4.56e-05 $\pm$ 2.8e-06 | 0.0015 $\pm$ 7e-05 |
| | Clipping ($\beta = 0.01$) | 0.0302 $\pm$ 0.0005 | 4.56e-05 $\pm$ 2.8e-06 | 0.0015 $\pm$ 7e-05 |
| | Clipping ($\beta = 0.1$) | 0.0302 $\pm$ 0.0005 | 4.56e-05 $\pm$ 2.81e-06 | 0.0015 $\pm$ 7e-05 |
| | Clipping ($\beta = 1.0$) | 0.1001 $\pm$ 0.0018 | 5.19e-05 $\pm$ 2.81e-06 | 0.0170 $\pm$ 0.0017 |

# C  Additional Experimental Details

## C.1  Calibration

Figure 4: Calibration of classifiers for density ratio estimation.

We found in all our cases that the binary classifiers used for training the model were highly calibrated by default and did not require any further recalibration. See for instance the calibration of the binary classifier used for goodness-of-fit experiments in Figure 4. We performed the analysis on a held-out set of real and generated samples and used 10 bins for computing calibration statistics.

We believe the default calibration behavior is largely due to the fact that our *binary* classifiers distinguishing real and fake data do not require very complex neural networks architectures and training tricks that lead to miscalibration for *multi-class* classification. As shown in [61], shallow networks are well-calibrated and [62] further argue that a major reason for miscalibration is the use of a softmax loss typical for multi-class problems.

## C.2  Synthetic experiment

The classifier used in this case is a multi-layer perceptron with a single hidden layer of 100 units and has been trained to minimize the cross-entropy loss by first order optimization methods. The dataset used for training the classifier consists of an equal number of samples (denoted as $n$ in Figure 1) drawn from the generative model and the data distribution.

## C.3  Goodness-of-fit testing

We used the Tensorflow implementation of Inception Network [63] to ensure the sample quality metrics are comparable with prior work. For a semantic evaluation of difference in sample quality, this test is performed in the feature space of a pretrained classifier, such as the prefinal activations of the Inception Net [64]. For example, the Inception score for a generative model $p_\theta$ given a classifier $d(\cdot)$ can be expressed as:

$$\text{IS} = \exp(\mathbb{E}_{\mathbf{x} \sim p_\theta}[\text{KL}(d(y|\mathbf{x}), d(y))]).$$

The FID score is another metric which unlike the Inception score also takes into account real data from $p_{\text{data}}$. Mathematically, the FID between sets $S$ and $R$ sampled from distributions $p_\theta$ and $p_{\text{data}}$ respectively, is defined as:

$$\text{FID}(S, R) = \|\mu_S - \mu_R\|_2^2 + \text{Tr}(\Sigma_S + \Sigma_R - 2\sqrt{\Sigma_S \Sigma_R})$$

where $(\mu_S, \Sigma_S)$ and $(\mu_R, \Sigma_R)$ are the empirical means and covariances computed based on $S$ and $R$ respectively. Here, $S$ and $R$ are sets of datapoints from $p_\theta$ and $p_{\text{data}}$. In a similar vein, KID compares statistics between samples in a feature space defined via a combination of kernels and a pretrained classifier. The standard kernel used is a radial-basis function kernel with a fixed bandwidth of $1$. As desired, the score is optimized when the data and model distributions match.

We used the open-sourced model implementations of PixelCNN++ [27] and SNGAN [11]. Following the observation by [38], we found that training a binary classifier on top of the feature space of any

pretrained image classifier was useful for removing the low-level artifacts in the generated images in classifying an image as real or fake. We hence learned a multi-layer perceptron (with a single hidden layer of 1000 units) on top of the 2048 dimensional feature space of the Inception Network. Learning was done using the Adam optimizer with the default hyperparameters with a learning rate of 0.001 and a batch size of 64. We observed relatively fast convergence for training the binary classifier (in less than 20 epochs) on both PixelCNN++ and SNGAN generated data and the best validation set accuracy across the first 20 epochs was used for final model selection.

### C.4 Data Augmentation

Our codebase was implemented using the PyTorch library [65]. We built on top of the open-source implementation of DAGAN[4] [1].

A DAGAN learns to augment data by training a conditional generative model $G_\theta : \mathcal{X} \times \mathcal{Z} \rightarrow \mathcal{X}$ based on a training dataset $D_{cl}$. This dataset is same as the one we used for training the generative model and the binary classifier for density ratio estimation. The generative model is learned via a minimax game with a critic. For any conditioning datapoint $\mathbf{x}_i \in D_{train}$ and noise vector $\mathbf{z} \sim p(\mathbf{z})$, the critic learns to distinguish the generated data $G_\theta(\mathbf{x}_i, \mathbf{z})$ paired along with $\mathbf{x}_i$ against another pair $(\mathbf{x}_i, \mathbf{x}_j)$. Here, the point $\mathbf{x}_j$ is chosen such that the points $\mathbf{x}_i$ and $\mathbf{x}_j$ have the same label in $D_{cl}$, i.e., $y_i = y_j$. Hence, the critic learns to classify pairs of (real, real) and (real, generated) points while encouraging the generated points to be of the same class as the point being conditioned on. For the generated data, the label $y$ is assumed to be the same as the class of the point that was used for generating the data. We refer the reader to [1] for further details.

Given a DAGAN model, we additionally require training a binary classifier for estimating importance weights and a multi-class classifier for subsequent classification. The architecture for both these use cases follows prior work in meta learning on Omniglot [66]. We train the DAGAN on the 1200 classes reserved for training in prior works. For each class, we consider a 15/5/5 split of the 20 examples for training, validation, and testing. Except for the final output layer, the architecture consists of 4 blocks of 3x3 convolutions and 64 filters, followed by batch normalization [64], a ReLU non-linearity and 2x2 max pooling. Learning was done for 100 epochs using the Adam optimizer with default parameters and a learning rate of 0.001 with a batch size of 32.

### C.5 Model-based Off-policy Policy Evaluation

For this set of experiments, we used Tensorflow [63] and OpenAI baselines[5] [67]. We evaluate over three envionments viz. Swimmer, HalfCheetah, and HumanoidStandup (Figure 5. Both HalfCheetah and Swimmer rewards the agent for gaining higher horizontal velocity; HumanoidStandup rewards the agent for gaining more height via standing up. In all three environments, the initial state distributions are obtained via adding small random perturbation around a certain state. The dimensions for state and action spaces are shown in Table 6.

(a) Swimmer      (b) HalfCheetah      (c) HumanoidStandup

Figure 5: Environments in OPE experiments.

Our policy network has two fully connected layers with 64 neurons and tanh activations for each layer, where as our transition model / classifier has three hidden layers of 500 neurons with swish activations [68]. We obtain our evaluation policy by training with PPO for 1M timesteps, and our behavior policy by training with PPO for 500k timesteps. Then we train the dynamics model $P_\theta$ for

Table 6: Statistics for the environments.

| Environment | State dimensionality | # Action dimensionality |
|---|---|---|
| Swimmer | 8 | 2 |
| HalfCheetah | 17 | 6 |
| HumanoidStandup | 376 | 17 |

Table 7: Off-policy policy evaluation on MuJoCo tasks. Standard error is over 10 Monte Carlo estimates where each estimate contains 100 randomly sampled trajectories. Here, we perform stepwise LFIW over transition triplets.

| Environment | $v(\pi_e)$ (Ground truth) | $\tilde{v}(\pi_e)$ | $\hat{v}(\pi_e)$ (w/ LFIW) | $\hat{v}_{80}(\pi_e)$ (w/ LFIW) |
|---|---|---|---|---|
| Swimmer | $36.7 \pm 0.1$ | $100.4 \pm 3.2$ | $19.4 \pm 4.3$ | $\mathbf{48.3} \pm 4.0$ |
| HalfCheetah | $241.7 \pm 3.6$ | $204.0 \pm 0.8$ | $\mathbf{229.1} \pm 4.9$ | $214.9 \pm 3.9$ |
| HumanoidStandup | $14170 \pm 5.3$ | $8417 \pm 28$ | $\mathbf{10612} \pm 794$ | $9950 \pm 640$ |

100k iterations with a batch size of 128. Our classifier is trained for 10k iterations with a batch size of 250, where we concatenate $(\mathbf{s}_t, \mathbf{a}_t, \mathbf{s}_{t+1})$ into a single vector.

Figure 6: Estimation error $\delta(v) = v(\pi_e) - \hat{v}_H(\pi_e)$ for different values of $H$ (minimum 0, maximum 100). Shaded area denotes standard error over different random seeds; each seed uses 100 sampled trajectories. Here, we use LFIW over transition triplets.

### C.5.1 Stepwise LFIW

Here, we consider performing LFIW over the transition triplets, where each transition triplet $(\mathbf{s}_t, \mathbf{a}_t, \mathbf{s}_{t+1})$ is assigned its own importance weight. This is in contrast to assigning a single importance weight for the entire trajectory, obtained by multiplying the importance weights of all transitions in the trajectory. The importance weight for a transition triplet is defined as:

$$\frac{p^\star(\mathbf{s}_t, \mathbf{a}_t, \mathbf{s}_{t+1})}{\tilde{p}(\mathbf{s}_t, \mathbf{a}_t, \mathbf{s}_{t+1})} \approx \hat{w}(\mathbf{s}_t, \mathbf{a}_t, \mathbf{s}_{t+1}), \tag{26}$$

so the corresponding LFIW estimator is given as

$$\hat{v}(\pi_e) = \mathbb{E}_{\tau \sim \tilde{p}(\tau)} \left[ \sum_{t=0}^{T-1} \hat{w}(\mathbf{s}_t, \mathbf{a}_t, \hat{\mathbf{s}}_{t+1}) \cdot r(\mathbf{s}_t, \mathbf{a}_t) \right]. \tag{27}$$

We describe this as the "stepwise" LFIW approach for off-policy policy evaluation. We perform self-normalization over the weights of each triplet.

From the results in Table 7 and Figure 6, stepwise LFIW also reduces bias for OPE compared to without LFIW. Compared to the "trajectory based" LFIW described in Eq. (20), the stepwise estimator has slightly higher variance and weaker performance for $H = 20, 40$, but outperforms the trajectory level estimators when $H = 100$ on HalfCheetah and HumanoidStandup environments.

## Footnotes

[4] `https://github.com/AntreasAntoniou/DAGAN.git`

[5] `https://github.com/openai/baselines.git`