[Reviews · NeurIPS 2019]

Reviewer 1



Originality and Significance: In light of submission with ID 6290, the method proposed in this paper is not novel. However the applications are different: submission 6290 focuses on learning generative models that are not influenced by the bias in the data whereas this submission focuses on reducing the mismatch between the trained generative model and the data distribution for Monte Carlo estimation at test time. Although the idea of importance weighting and estimating density ratios with a classifier is not new I believe the applications in the two papers are potentially useful. Quality and Clarity: The paper is well written. The related work section is good but there is a missing line of related work that was not mentioned, that of using importance weighting in the context of variational inference. See for example [1] and many follow-up papers. The experiments section is nicely done. Questions and Minor Comments: 1--how do you ensure that the generative distribution p_{\theta} has bigger support than p? 2--Is there a way to measure the bias introduced by p_{\theta}? What is this bias? 3--How did you deal with high variance in importance reweighting? What usually happens in importance weighting is that all the components of the importance weights collapse to 0 except for one component which eats all the mass...This happens even with your proposed trick on line 135 (self-normalization) and I can imagine this would also happen for the solution on line 137 (flattening). Clipping (as proposed on line 141) might be promising in practice however you do lose the asymptotic unbiasedness of importance weighting in this case. Please clarify. 4--how did you choose the clipping threshold \beta? [1] Importance-Weighted Autoencoders. Burda et al., 2015.

Reviewer 2



=Summary= The paper propose a method for correcting the bias in the outcomes of pretrained deep generative models. Given data from a generator distribution and the real distribution, the paper uses importance reweighting to up/down-weigh the generated samples. The importance weights are computed using a probabilistic binary classifier that predicts the identity of the data distribution. Experiments are shown on several tasks to show that the importance reweighting improves the task performance. ---- =Originality= Medium. The importance weighting using binary classification is a well-known technique. However, its usage in pretrained generative models is interesting. =Quality= Medium. The experimental section is well-written and the paper clearly points out potential drawbacks. =Clarity= Low-Medium. The paper is in general easy to read. However, some design choices can be explained better. Please see detailed comments below. =Significance= Medium. The proposed scheme can be a good addition to a deep generative model practitioner's toolkit. The framework has some stark limitations (e.g., the requirement regarding joint support of the real and generated data), as also pointed out in the paper. But it is still a useful addition to the growing literature on deep generative modeling research. ---- =Detailed comments and suggestions= - Lines 135-143: It is not clear what the corrective measures proposed here do on an intuitive and theoretical level. For examples, given very high and very low importance weights in a minibatch (corresponding to real and generated data), how does normalization help in terms of obtaining the "true" importance weights? - Similarly, are there any guidelines on when to apply each of the above corrective measures and how should the hyperparameters be selected? - Line 161: It is a bit surprising to see that the binary classifiers are calibrated by default, given that deep models are known to be very prone to miscalibration (https://arxiv.org/pdf/1706.04599.pdf). How precisely is miscalibration measured (e.g., via expected calibration error as described in the reference above)? - Line 168: For computing the reference scores in Table 1, how precisely is the real data split? Is it a 50-50 split? - Line 129, 304: The reviewer appreciates the fact that the paper is quite open about the potential limitations in the proposed methodology. - One limitation that is worth mentioning is that the proposed method cannot be used to generate new unbiased samples. - A question inspired by the closely related work of [45]: Ignoring the mode-dropping phenomenon of GANs (that is, assuming that the real and generated distributions have joint support), would the proposed importance weighing mechanism be obsolete if one were to train the GAN for a very long time? ------- = Update after the rebuttal = Most of my questions were addresses in the rebuttal. It is good to see the plot showing that the models are already well-calibrated. It would perhaps be helpful to add the plot (or at least the reference by Danescu-Mizil and Caruana) in the final version so that the readers are aware of 1) the potential for miscalibration and 2) the fact that these models are in fact well-calibrated.

Reviewer 3



The paper proposes an importance weights based scheme to correct the bias in the distribution learned by a generative model. Specifically, the authors employ a classifier to distinguish between a true data sample and a sample from the generative model. This binary classifier is used to estimate the likelihood ratio when the true distribution is unknown and the model distribution is intractable. Authors also discuss practical techniques to address the issue of imperfect classifier and reduce the variance of the Monte Carlo estimates. There include normalizing the weights across a batch of data samples, smoothing as well as clipping. Standard metrics to evaluate sample quality of the generative models such as FID and Inception score show improvement when the Monte Carlo estimates using the learned importance weights are used. This also appears to help on downstream domain adaptation task on Omniglot task. Questions/Concerns: It is not clear how the learned classifier is calibrated. The calibration seems quite crucial and this is often difficult, especially for deep neural network classifier trained on high dimensional data. The post-hoc normalization schemes are not well motivated. How do these interact with calibration? p, p_data, p_theta needs to defined early on. It is nor clear if p_mix, eq (2) and related discussion add much to the discussion. Why does D_g + LFIW not show much improvement over D_g. In the toy experiment, how do the results shown in Figure 1 change as the two modes in the true distribution get closer and closer? Domain adaptation on standard benchmark datasets and comparison with baselines will strengthen the empirical results. Typo: Policy evaluation numbers are missing in the introduction. Update: Thanks for answering most of my questions and of other reviewers. I have raised my score.

[Author Response · NeurIPS 2019]



We thank all reviewers for their helpful and detailed comments! We have addressed the issue of dual submission in detail in the rebuttal of paper #6290. As R1 notes and we further elucidate, the problem setting, algorithm specifics, and use-case scenarios of the two papers are different and independent – **model bias** of a **pretrained model** for downstream **Monte Carlo evaluation** here vs. **data bias** during **weakly-supervised learning** for **fair data generation** in #6290.

• **R1:** *Support of the generative distribution $p_\theta$ and $p$.* Our meta-algorithm takes as input a learned model $p_\theta$ and $p$ so satisfying the support assumption is tied to the *training* of $p_\theta$ (which we do not consider in this work). Nevertheless for a likelihood-based model, the support assumption can be empirically verified via evaluating $p_\theta$ on held-out data. The assumption holds true for most variants of VAEs, flows, and autoregressive which have full support by design. We also consider a more general case where we have only sample access to both $p_\theta$ and $p$, where estimating the support is a computationally hard problem (related to estimating the entropy of arbitrary distribution via samples).

To address issues related to the estimation of importance weights via a learned classifier, tricks such as perturbations via small, random Gaussian noise (which has full support), regularization (dropout, early stopping etc.) during training (L306-307), as well as post-processing schemes (L135-143) can be applied. Empirically, we find self-normalization along with early stopping during training (based on validation data) to be sufficient for ensuring good downstream performance for various generative models (GANs, autoregressive models) and modalities considered in this work.

• **R1:** *Defining and measuring the bias introduced by $p_\theta$.* In this work, bias is defined w.r.t. any function $f$ defined over the data domain. Given $p_{\text{data}}$, $p_\theta$ and $f$, the bias is defined as the difference in the expected value of $f$ with respect to $p_{\text{data}}$ and $p_\theta$ (Footnote 1, Page 1). When $p_{\text{data}}$ and $p_\theta$ are not known directly, the bias can be estimated empirically via Monte Carlo using a sufficiently large number of samples from $p_\theta$ and $p_{\text{data}}$ e.g., as shown in Table 1 and Appendix B.

• **R1:** *High variance in importance reweighting.* As with other applications of importance weighting, the extent of and solutions to the high variance issue are empirically motivated. They could introduce a bias (e.g., clipping) but reduce variance more favorably in the tradeoff. In our setting, the primary limitation was that the estimated importance weights could all be small due to artifacts in the generations that were easy to detect via the binary classifier. While we found self-normalization to be most effective, we note in L142 that schemes for post-processing importance weights could be potentially combined, e.g., self-normalized weights could be clipped when variance is a larger issue.

• **R1:** *Choosing the clipping threshold $\beta$.* We consider $\beta$ as a validation hyperparameter with values in {0.001, 0.01, 0.01, 1} chosen to maximally reduce the bias in Monte Carlo evaluation of a downstream function of interest.

• **R2:** *Intuition and guidelines for design choices in L135-143.* Self-normalization is applied only for the generated samples (i.e., those that contribute to bias in Monte Carlo evaluation). Like with other applications, the usage is empirically driven. Generative models tend to produce artifacts that are easy to detect via classifiers and hence, the estimated importance weights are very small ($\ll 1$). In all our experiments, self-normalization was essential to circumvent this issue (see expts. in Tables 4, 5 in Appendix where self-normalization leads to a 53% improvement in mean squared error over vanilla importance weighting). It is hyperparameter free and easy to apply. If variance is high, the range of the weights can be restricted via clipping or flattening with hyperparamters $\beta, \alpha$ tuned on validation set.

• **R2:** *Data split for reference scores in L168.* Yes, the split is 50-50.

• **R2:** *Running procedure in [45] for long.* Yes, ignoring the high computational requirements of [45] and the fact that the upper bound for rejection sampling is a heuristic estimate, the procedure in [45] could achieve the same effect as the proposed importance weighting approach.

• **R2, R3:** *Calibration.* We believe the default calibration behavior is largely due to the fact that our **binary** classifiers distinguishing real and fake data do not require very complex neural networks architectures and training tricks that lead to miscalibration for **multi-class classification**. As shown by Niculescu-Mizil & Caruana (2005), shallow networks are well-calibrated and Guo et al. (2017) further argue that a major reason for miscalibration is the use of a softmax loss typical for multi-class problems. Top-left figure shows example calibration curves for the experiment in 5.1.

• **R3:** *Interaction of post-hoc normalization schemes with calibration.* While calibration is necessary for a sound density ratio estimation procedure, the utility of the derived importance weights for downstream tasks depends on the underlying expectation of interest. These expectations are evaluated with finite samples and hence, the asymptotic properties of importance weighting (e.g., unbiasedness) are traded off for improved downstream performance using self-normalization and other post-processing schemes.

• **R3:** *Domain adaptation.* We clarify that we are considering the task of multi-class classification and not domain adaptation (L179-181). As we note in L182-183, the Omniglot dataset is a particularly relevant test bed for data augmentation since there are a large number of classes and a few number of training examples per class. We will consider other related scenarios in future version!

• **R3:** $D_{\text{g}}$ + LFIW *vs.* $D_{\text{g}}$. Note that this experiment does not only involve Monte Carlo evaluation of a supervised loss but also optimization via gradient methods. In the absence of real data $D_{\text{cl}}$, the classifier training is dominated by $D_{\text{g}}$ and correcting the bias in the dataset via LFIW towards an unseen dataset ($D_{\text{cl}}$) can potentially have limited gains.

• **R3:** *Modes getting closer in Fig 1.* As modes get closer, the importance weights will approach 1 (and the class probabilities will approach 0.5) since the mismatch in generative model and data distributions will accordingly decrease.

[Meta-Review · NeurIPS 2019]

Congratulations, your paper has been accepted for publication at NeurIPS2019. The reviewers found it to be a novel well executed piece of work. When preparing the camera ready version, please bear in mind the reviewers comments. In particular - Please carefully define what bias is. The footnote on p1 is somewhat vague. Is a generative model biased if it is biased for *any* statistic, or just for some set of minimal statistics? - It would be helpful to add the plot showing that the models are already well-calibrated (or at least the reference by Danescu-Mizil and Caruana) in the final version so that the readers are aware of 1) the potential for miscalibration and 2) the fact that these models are in fact well-calibrated.